# CRC-Aided Adaptive BP Decoding of PAC Codes

**DOI:** 10.3390/e24081170

**Published:** 2022-08-22

**Authors:** Xianwen Zhang, Ming Jiang, Mingyang Zhu, Kailin Liu, Chunming Zhao

**Affiliations:** 1The National Mobile Communications Research Laboratory, Southeast University, Nanjing 210096, China; 2Purple Mountain Laboratories, Nanjing 211111, China

**Keywords:** PAC codes, CRC-aided, list decoding, adaptive belief propagation

## Abstract

Although long polar codes with successive cancellation decoding can asymptotically achieve channel capacity, the performance of short blocklength polar codes is far from optimal. Recently, Arıkan proposed employing a convolutional pre-transformation before the polarization network, called polarization-adjusted convolutional (PAC) codes. In this paper, we focus on improving the performance of short PAC codes concatenated with a cyclic redundancy check (CRC) outer code, CRC-PAC codes, since error detection capability is essential in practical applications, such as the polar coding scheme for the control channel. We propose an enhanced adaptive belief propagation (ABP) decoding algorithm with the assistance of CRC bits for PAC codes. We also derive joint parity-check matrices of CRC-PAC codes suitable for iterative BP decoding. The proposed CRC-aided ABP (CA-ABP) decoding can effectively improve error performance when partial CRC bits are used in the decoding. Meanwhile, the error detection ability can still be guaranteed by the remaining CRC bits and adaptive decoding parameters. Moreover, compared with the conventional CRC-aided list (CA-List) decoding, our proposed scheme can significantly reduce computational complexity, to achieve a better trade-off between the performance and complexity for short PAC codes.

## 1. Introduction

Polar codes are the first channel codes that are theoretically proven to achieve the capacity of any symmetrical binary input discrete memoryless channel [1]. However, the error correction performance of finite-length polar codes is far from optimal. A practical method to solve this problem is cyclic redundancy check (CRC) precoding, followed by successive cancellation list (SCL) decoding, which is called CA-SCL decoding [2]. Recently, Arikan proposed polarization-adjusted convolutional (PAC) codes [3], which used a convolutional pre-transformation before the polarization network, effectively alleviating the capacity loss caused by the slow polarization of the short and medium length channels. Although the sequential decoding algorithm used in this paper has high complexity, the performance of PAC (128, 64) code is almost identical to the PPV bound [4]. Using list decoding and sequential decoding, PAC codes always show better performance than polar codes with the same blocklength [5]. However, in some scenarios (for example the control channel in 5G NR), it is very critical to detect the decoding error and avoid missed detection. Therefore, it causes a severe difficulty for the practical applications of PAC codes. The typical approach is to add CRC redundant bits at the end of the information bits before PAC encoding, forming a CRC-PAC concatenated coding scheme.

The paper [6] proves that convolutional pre-transformation does not decrease the minimum distance of codes, but will substantially reduce the number of codewords with the minimum distance. In addition, there are some attempts to concatenate CRC codes and PAC codes to improve decoding performance [7], where the application of the CRC-aided list (CA-List) decoding algorithm to CRC-PAC codes is further discussed. Simulation results show that the CRC-PAC codes can also approach the random-coding union (RCU) bounds for (256, 128) and (512, 256) codes. However, the aforementioned concatenated scheme only uses several relative short CRC codes and does not consider the undetected error rate (UER), which is a particularly important indicator that determines whether the decoded output is a valid codeword. According to the latest 5G specification (release 17), the maximum length of CRC bits attached to polar codes is 24 [8], providing a flexible approach to trade off the error correction and detection capabilities.

Jiang and Narayanan proposed the adaptive belief propagation (ABP) algorithm with polynomial complexity for Reed–Solomon (RS) codes [9]. The ABP algorithm has the best performance among all known decoding algorithms of RS codes. Its basic idea is to continuously use Gaussian elimination to adjust the binary image parity-check matrix according to the bit reliabilities, by diagonalizing the sub-matrix associated with the less reliable decision bits. On the one hand, it allows the less reliable bits to get more assistance when updating soft information, and on the other hand, it minimizes the error propagation of these bits in message passing decoding. The ABP algorithm uses a Gaussian elimination series with very high complexity, but it provides quite impressive performance almost close to the maximum likelihood (ML) lower bound, which is significantly better than other exponential-complexity algorithms, including the Kotter–Vardy algorithm and ordered statistic decoding (OSD). Although ABP is mainly used for soft-decision decoding of RS codes, in fact, it can be employed for almost all short blocklength codes, including short low-density parity-check (LDPC) codes and polar codes [10,11]. Besides, for LDPC codes and polar codes in 5G NR, there is already a relatively efficient implementation scheme in hardware [12,13,14]. The paper [13] proposed an efficient implementation of QC-LDPC codes for 5G NR by pruning the full-base matrix before use, improving the throughput of QC-LDPC codes in 5G NR. Therefore, the hardware implementation is also feasible for PAC codes.

This paper mainly concentrates on PAC codes with short blocklength and proposes a partial CRC-aided ABP algorithm as a powerful substitute to CA-List decoding. The contributions of this paper can be summarized in the following three aspects:We derive the parity-check matrices of PAC codes suitable for message passing decoding.We propose a partial CRC-aided ABP algorithm by designing the joint parity-check matrices of CRC-PAC codes. The ABP algorithm clearly supports the use of partial CRC bits by combining some parity-check constraints of a CRC code and the parity-check matrix of a PAC code.We analyze and compare the performance of our proposed algorithm and the conventional CA-List decoding for punctured PAC codes. The performance influence of puncturing is taken into consideration.

Simulation results show that our method achieves a large gain over the CA-List decoding of high rate or punctured CRC-PAC codes while keeping some error detection capability.

## 2. The Structures of PAC Codes and Concatenated CRC-PAC Codes

### 2.1. The Structures of PAC Codes

According to Arikan’s conclusions, PAC codes and polar codes share the same channel polarization theory. The code length *N* of PAC codes is an exponent of 2, that is $N=2^{n}$. The length of information bits *K* can be any integer from 1 to *N*. The information sequence $d=(d_{0},d_{1},...,d_{K-1})$ is then mapped to the bit sequence $v=(v_{0},v_{1},...,v_{N-1})$ by rate profiling. The index set I contains the *K* most reliable positions of v. The *K* information bits are filled to the positions of the data index set I. The rest index set F represents the indexes of frozen positions, which are filled with zeros. The PAC code x is obtained by multiplying the two transfer matrices T and $P_{n}$
(1)$$x=v\cdot T\cdot P_{n},$$
where T and $P_{n}$ correspond to the transformations of convolution and polarization, respectively. Generally, the transfer matrix T is an upper-triangular Toeplitz matrix, which can be represented by the first row, the impulse response $\mathrm{IR}=(IR_{0},IR_{1},...,IR_{m})$ of convolution operation. The polarization matrix is denoted by $P_{n}=F^{⨂n}$, where $F^{⨂n}$ is the *n*-th Kronecker power of $F=\left[\begin{array}{cc} 1 & 0\\ 1 & 1 \end{array}\right]$ with $n=\log_{2}N$. The performance of PAC codes is highly sensitive to the choice of the data index set, and the impact of the impulse response IR is relatively small. To select the data index set, a valid criterion is the RM evaluation s(i)=w(i−1), where w(i−1) is the weight in the binary representation of i−1 and satisfies 1≤i≤N. We can choose the *i* corresponding to the largest *K* values among s(1),s(2),...,s(N) to form the data index set. Furthermore, the Bhattacharyya parameter construction method [15] is used to select the data indexes from those with the same weight of binary representations for better performance.

The performance of the optimized PAC (128, 96) code is shown in Figure 1. Among them, the information bit selection of polar codes is still in accordance with the 5G standard, adding 8 CRC redundant bits at the frozen positions, and using the CA-SCL algorithm. Similarly, the PAC code also uses the list decoding, and the list sizes for both codes are set to 32. In Figure 1, the gain of the PAC code before optimized by the Bhattacharyya parameter construction is less than 0.05 dB compared to the corresponding polar code. But after optimization, the PAC code obtains a gain of about 0.15 dB. It can be seen that using the Bhattacharyya parameter method as an auxiliary condition for the RM criterion can significantly improve the decoding performance of PAC codes whose length of information bits is not located at the demarcation point.

### 2.2. The Structures of CRC-PAC Concatenated Codes

Throughout this paper, the code blocklength, the length of information bits and the code rate of a CRC-PAC code are denoted by *N*, *A*, and R=A/N, respectively. Referring to 5G standards, a typical concatenated CRC-PAC code includes three parts, information bits of length *A*, the CRC bits of length L=K−A, and M=N−K parity-check bits of the PAC code. While, a CRC code usually can be described by a non-systematic and high-density generator matrix with a generator polynomial $g\left(x\right)=g_{L}x^{L}+g_{L-1}x^{L-1}\cdots +g_{0}$, where the value of $g_{i}$, i∈⟦0,L⟧ is 0 or 1. According to the characteristics of cyclic codes, the generator matrix can be obtained by cyclic shifting of the generator polynomial
(2)$$G_{CRC}=\left(\begin{array}{cccccc} g_{L} & g_{L-1} & \cdots & g_{0}\\ \; & \ddots & \ddots & \ddots & \ddots \\ \; & \; & g_{L} & g_{L-1} & \cdots & g_{0} \end{array}\right).$$ We apply Gaussian elimination to convert the generator matrix of the CRC code into the systematic form
(3)$$G_{S}=\left[I_{\left(A\times A\right)}|J_{\left(A\times L\right)}\right],$$
where $I_{\left(A\times A\right)}$ represents an identity matrix with size *A*. According to the relationship between the generator matrix and the parity-check matrix, it is easy to get the corresponding systematic parity-check matrix of the CRC code as
(4)$$H_{S}=\left[{\left(J_{\left(A\times L\right)}\right)}^{T}|I_{\left(L\times L\right)}\right].$$

Although *L* redundant parity bits are used for error detection, there is still a certain possibility of missed detections. Missed detection is more important to the real-time services in high signal-to-noise ratio (SNR) regimes. According to [16], we also evaluate the performance by the UER criterion, which is defined as the ratio of the number of miss-detected blocks to the number of all decoded blocks. In general, the UER performance is closely related to the length of CRC parity-check bits *L* and frame error rate (FER), and its upper limit [16] is
(5)$$\mathrm{UER}\approx \frac{1}{2^{L}}\times \mathrm{FER}.$$

The UER performance is a critical factor for the CRC-aided decoding algorithm, which incorporates all or partial CRC redundant bits to enhance error correction capabilities. Put another way, it gives up all or partial error detection capability in exchange for improving decoding performance.

A concatenated CRC-PAC code can be regarded as encoding the original information bits with the joint generator matrix, which is the multiplication of two generator matrices of a CRC code and a PAC code. In order to better explain the concatenated structure of CRC-PAC codes, in the following discussion, PAC codes are denoted as inner codes, and the corresponding generator matrices (parity-check matrices) are identified by the subscript “*I*”, namely $G_{I}$$\left(H_{I}\right)$. The generator matrix of a PAC code can be expressed as
(6)$$G_{I}=T\cdot P_{n}.$$

Similarly, CRC codes are defined as outer codes and the corresponding matrices use the subscript “*O*” to be identified, namely $G_{O}$$\left(H_{O}\right)$. The original information bits with length *A*, the outer coded block with CRC redundant bits and the inner PAC coded block are marked as $m_{A}$, m, and c, respectively. Let φ(b) and φ(B) denote the operations to add zero bits and zero columns to a row vector b and a matrix B according to the frozen positions in the set F, respectively. The encoding process of the concatenated CRC-PAC codes can be expressed as
(7)$$c=\varphi \left(m_{A}\cdot G_{O}\right)\cdot G_{I}.$$ We notice that $\varphi \left(m_{A}\cdot G_{O}\right)=\varphi \left(m_{A}\right)\cdot \varphi \left(G_{O}\right)$, so we define the joint generator matrix
(8)$$G=\varphi \left(G_{O}\right)\cdot G_{I}.$$

## 3. CRC-Aided ABP Decoding

### 3.1. The Parity-Check Matrices of CRC-PAC Concatenated Codes

In this subsection, we derive the parity-check matrix of the PAC code, and design the joint parity check matrix by integrating some parity-check constraints of the CRC code into the parity-check matrix of the PAC code. Similar to the derivation process of the parity-check matrix of a polar code which is formed from the columns of $P_{n}$ with the frozen indexes F [17], we can get the parity-check matrix of a PAC code. Since the codeword of a PAC code is given by $x=v\cdot T\cdot P_{n}$, the transformation can be inverted as
(9)$$v=x\cdot {P_{n}}^{-1}\cdot T^{-1}.$$

Note that for the generator matrix of a polar code, $P_{n}$ is directly equal to ${P_{n}}^{-1}$ in the binary field. Then, we have
(10)$$v=x\cdot P_{n}\cdot T^{-1}.$$

Hence, setting indices $v_{i}=0$ for all i∈F selects the corresponding columns in $P_{n}\cdot T^{-1}$ imposing parity-check constraints on the codeword x, and we can get the parity-check matrix of a PAC code $H_{I}$. A CRC code satisfies the following parity-check constraints,
(11)$$m\cdot {H_{O}}^{T}=0.$$ Considering the encoding process of CRC-PAC codes, we have
(12)$$c=\varphi \left(m_{A}\cdot G_{O}\right)\cdot T\cdot P_{n}.$$ For CRC codes, we can get
(13)$$m_{A}\cdot G_{O}\cdot {H_{O}}^{T}=0.$$ Then, the computation for (13) can be rewritten as
(14)$$\varphi \left(m_{A}\cdot G_{O}\right)\cdot T\cdot P_{n}\cdot P_{n}\cdot T^{-1}\varphi \left({H_{O}}^{T}\right)=0,$$
where $\varphi \left(m_{A}\cdot G_{O}\right)\cdot T\cdot P_{n}$ is the coded block of a PAC code and satisfies the modified parity-check matrix
(15)$${\hat{H}}_{O}={\left(P_{n}\cdot T^{-1}\varphi \left({H_{O}}^{T}\right)\right)}^{T}.$$

Further, considering the parity-check matrix of the PAC code, the joint parity-check matrix of the CRC-PAC code $\hat{H}$ can be defined as
(16)$$\hat{H}=\left[\begin{array}{c} H_{I}\\ {\left(P_{n}\cdot T^{-1}\cdot \varphi \left({H_{O}}^{T}\right)\right)}^{T} \end{array}\right].$$

Similarly, we can get a partial CRC-aided joint parity-check matrix
(17)$${\hat{H}}_{p}=\left[\begin{array}{c} H_{I}\\ {\tilde{H}}_{O}. \end{array}\right]$$
where ${\tilde{H}}_{O}$ represents the first $\tilde{L}$ rows of matrix ${\hat{H}}_{O}$. When all CRC bits are used for error correction, ${\hat{H}}_{O}={\tilde{H}}_{O}$. Using such a partial CRC-aided joint parity-check matrix, we can perform the partial CRC-aided ABP (CA-ABP) algorithm on the CRC-PAC concatenated codes. By combining the partial parity-check matrix of the CRC code with that of the PAC code, the joint parity-check matrix demonstrates better linear random features than that of the PAC code, which is more beneficial to the ABP decoding of CRC-PAC codes with better distance spectrum.

### 3.2. Decoding Steps of CA-ABP Algorithm

The parity-check matrix of a PAC code is not only highly density, but also contains a large number of cycles of length four that severely degrade the performance of iterative decoding. Therefore, the direct use of BP decoding will be heavily influenced by error propagation, and the performance of the conventional BP decoding on PAC codes is far away from that of the CA-List decoding algorithm. Whereas, the ABP decoding can be carried out effectively with a high density matrix, such as the binary image parity check matrix of an RS code. As for the ABP decoding of CRC-PAC codes, the joint parity-check matrix is initialized by (17) with the combination of $H_{I}$ and all or partial rows of the matrix ${\hat{H}}_{O}$. Channel outputs can provide reliable information for the sorting and Gaussian elimination in ABP scheme. Besides, if the ABP decoding successfully outputs a result, the remaining CRC bits can be used for data checking to avoid missed detection.

In addition, unlike the conventional ABP [9] method, our CA-ABP algorithm supports the use of partial CRC bits by combining some parity-check constraints of the CRC code and the parity-check matrix of the PAC code derived in the last subsection. Furthermore, the remaining CRC bits can be used for detection to lower UER.

The ABP decoding processes the reliability information through the following steps, including sorting, Gaussian elimination, and BP decoding. These key processes are listed in Algorithm 1. Here, the matrix used for Gaussian elimination is the joint parity-check matrix ${\hat{H}}_{p}$ of a CRC-PAC code. In this way, the participation of $\tilde{L}$ CRC bits may have considerable effect on the ABP algorithm, especially for short PAC codes, where the CRC redundancy accounts for a considerable proportion, and the advantages of the ABP algorithm on the joint parity check matrix will be more obvious.
**Algorithm 1:** ABP algorithm with partial CRC aided, CA-ABP($T_{I}$, $T_{O}$, $\tilde{L}$)
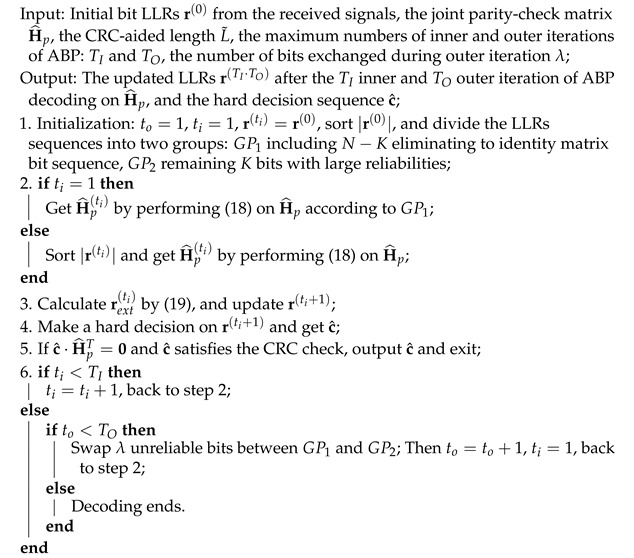


In the *j*-th iteration of the ABP algorithm, we perform Gaussian elimination on the reordered matrix ${\hat{H}}_{p}$ according to the reliability information, the magnitudes of log-likelihood ratios (LLRs) $r^{j}=\left(r_{1}^{j},r_{2}^{j},...,r_{N}^{j}\right)$, where the columns associated with the less reliable bits are transformed into the identity matrix form. This process is described as follows
(18)$${\hat{H}}_{p}^{\left(j\right)}=\phi ({\hat{H}}_{p},|r^{\left(j\right)}\left|\right).$$ In the stage of updating bit reliabilities, we calculate the sum of extrinsic reliability information $r_{ext}^{\left(j\right)}$ based on the LLRs of channel outputs $r^{\left(j\right)}$ and the matrix ${\hat{H}}_{p}^{\left(j\right)}$ after Gaussian elimination [9]. Then, we use the extrinsic information to modify the bit reliabilities, so as to realize the message passing and reliability updating
(19)$$r_{ext}^{\left(j\right)}=\psi \left({\hat{H}}_{p}^{\left(j\right)},r^{\left(j\right)}\right).$$
(20)$$r^{\left(j+1\right)}=r^{\left(0\right)}+\alpha r_{ext}^{\left(j\right)},$$
where the damping factor α, 0<α≤1, controls the updating rate of bit reliability information. In the practical implementation, the damping factor can be optimized by simulation. A variation that can help further improve performance is to run the ABP algorithm multiple times with the same initial LLRs from the channel outputs, but with the less reliable bits grouped differently [9]. The number of less reliable bits used for exchanging in the outer iteration for different groups is set to λ.

### 3.3. Decoding Complexity

The complexity of the CA-ABP algorithm mainly lies in the Gaussian elimination operations in the iterative process. It is a highly complex algorithm and can only be realized in a serial manner. One Gaussian elimination requires $N\times \min{\left\{K,(N-K)\right\}}^{2}$ binary operations. If the maximum numbers of outer and inner iterations are set to $T_{O}$ and $T_{I}$, respectively, the complexity of the ABP algorithm is approximately $O(T_{I}\cdot T_{O}\cdot N^{3})$. However, the parity-check matrix and CRC detection can both be used for early termination to lower the average complexity of the CA-ABP algorithm.

Due to early stopping used in our proposed scheme, in terms of complexity, the CA-ABP is more competitive than the CA-List decoding in the high SNR regime. In general, the computational complexity can be calculated by comparisons, additions, and multiplications. Most of these operations in the binary decoding algorithm can correspond to one or more equivalent additions, which are taken as the indicator in this paper. The detailed results are shown in Table 1, where η, $\epsilon =(T_{O}-1)\cdot \lambda$, and $T_{av}$, respectively, denote the list size of the CA-List algorithm, the total number of exchanged bits in outer iterations and the average number of iterations of CA-ABP, and other symbols can refer to the previous section of our paper.

In a bid to compare with CA-List algorithm, a short CRC-APC code of length N=128, CRC length L=24, and code rate R=40/128 is selected. The computational complexity comparisons of different algorithms are shown in Figure 2. With the SNR increasing, the computational complexity of the CA-ABP(20, 10, 16) decoding will be obviously lower than that of the CA-List(32) decoding. Further, the CA-List(1024) decoding requires 100∼1000 times more operations than our proposed algorithm in all the SNR range.

Here, we need to note that the decoding complexity of the proposed CA-ABP algorithm is quite different from that of the conventional ABP decoding due to the participation of partial CRC bits in error correction and the lower density of the joint parity-check matrices. More precisely, the complexity of the CA-ABP varies with the number of CRC bits participating in decoding. When all the CRC bits participate in the decoding, the complexity is maximized. While only partial CRC bits participating in the ABP decoding, the average decoding complexity will be significantly reduced by the early stopping with the joint parity-check matrices. Moreover, the binary parity-check matrix of the RS code expanded from the Vandermonde matrix over GF(q) usually has a density of near 50%, while our derived joint parity-check matrix of the CRC-PAC code is less than half of that of the RS code [11]. The proposed CA-ABP decoding on the parity-check matrix with lower density means fewer iterations than conventional ABP decoding, exhibiting lower complexity.

Although the implementation complexity of CA-ABP decoding increases sharply with the code length, there is already a relatively low-complexity hardware implementation for the Gaussian elimination of short codes [18]. Therefore, for short CRC-PAC codes, it is reasonable and practical to apply the CA-ABP decoding on the joint parity-check matrix ${\hat{H}}_{p}$.

## 4. Punctured PAC Codes

In this section, we consider the ABP decoding of punctured PAC codes. In the original construction, the blocklength *N* of a PAC code is strictly limited to an exponent of 2, and the number of information bits *K* is arbitrary. This blocklength limitation is a major disadvantage of PAC codes, which must be overcome in the practical application of PAC codes. The mapping from an information block of length *K* to an encoded block of length *E* is called the rate-compatible puncturing of PAC codes. Consider a punctured PAC code of (E,K). Puncturing is to modify the initial PAC code of (N,K) into a shortened code block of (E,K), where E=N−P, and P<N−K. That is to say, the number of punctured bits *P* cannot exceed the number of frozen bits. Obviously, the punctured bits are not transmitted after puncturing, and the code rate naturally increases.

There are many technical works on different puncturing methods for polar codes [19,20]. The paper [21] proposed a low-complexity puncturing scheme with a punctured set P denoted by
(21)$$P=B_{N}\left(0:P-1\right),$$
where $B_{N}\left(x\right)$ is the bit inversion function. Before transmission, *P* bits in the set P are selected to be punctured. According to the punctured positions determined by this method, it is obvious that the most unreliable decision bits are also in the set P. That is to say, before the list decoder, the soft information bits at the positions in set P is 0. Therefore, these positions can be set as frozen bits in the transmitter. The remaining frozen bits can still be selected by the construction algorithm according to the reliabilities of different positions of the mother code. Assume that the $R_{N}$ is sorted by the reliable indexes determined by the construction algorithm, and the elements of the set $R_{N}$ existing in the set P are excluded. Then, the *K* most reliable positions in the set $R_{N}$ are selected as the information bits, and the remaining positions are frozen bits. Due to the features of the RM construction, we select the Bhattacharyya parameter construction [1] to subdivide the indexes of positions corresponding to the same row weight. Figure 3 shows the performance of punctured PAC codes decoded by the list algorithm with the list size equals to 32. The blocklength *E* of the transmitted bits is fixed, and the FER performances with different lengths of information bits are compared, where the code length of the mother code is N=256.

It is obvious that as the code rate increases, the performance of punctured PAC codes worsens. The possible reason is that when the code rate increases, the relationship between the encoded bits is tighter, and the soft information is more important during decoding. So, the effect of puncturing and initializing the LLRs of untransmitted bits to 0 is greater on decoding. However, considering BP-like algorithms can better recover the original information bits of LDPC codes, we can further explore the performance of CA-ABP in punctured CRC-PAC codes.

## 5. Simulation Results

According to 5G standards, the UER of decoding should be under 1% of the FER [8]. In our proposed decoding, when CA-ABP decoding uses all CRC bits to assist in error correction, the UER may exceed the threshold requirement. There are several alternative methods to guarantee the UER performance. One is to reduce the number of CRC bits used in CA-ABP, and the other is to adjust the damping factor in CA-ABP algorithm. We can also use the metric threshold of Euclidean distance to lower the UER [22].

To the best of our knowledge, the CA-List decoding is the most efficient and popular algorithm for polar or PAC codes. Here, we compare the decoding performances of short CRC-PAC codes using CA-List and CA-ABP algorithms by simulation. For the purpose of evaluation, the partial CRC-aided parity-check matrices of PAC codes come from (17). The encoded bits are transmitted over the additive white Gaussian noise (AWGN) channel using the binary phase shift keying (BPSK) modulation. The receiver chooses CA-List decoding as the reference algorithm for PAC codes, the list sizes are respectively set to 32, 1024 and 2048, and at least 100 decoding errors are counted at each $E_{b}/N_{0}$. The UER performance are also evaluated since partial CRC bits are taken into error correction, thus we collect at least 50 undetected errors which satisfy all parity checks of PAC codes and the residual CRC detection in the CA-ABP decoding at each $E_{b}/N_{0}$ point.

For the CA-ABP algorithm, we set the maximum numbers of inner iterations and outer iterations to $T_{I}$ and $T_{O}$, respectively. The CRC-PAC encoding uses the concatenated structure in (7). For all CRC-PAC codes used in our simulations, the length of CRC bits is set to 24. The number of unreliable bits using for exchanging in outer iteration of ABP decoding is set to λ=1. In practice, when the code length increases, the value of λ should also increase, but it has little impact on the simulation results. The damping factor in our proposed CA-ABP algorithm is set to 0.08.

In order to deeply analyze the balance between UER and FER, we enable the CA-ABP algorithm to respectively use 14, 16, 18, and 20 CRC bits for auxiliary decoding of the PAC(128, 96) code. As indicated by the results, shown in the Figure 4, it is not surprising that as the number of auxiliary CRC bits in CA-ABP decreases, the FER performance of CA-ABP gradually degrades but the UER performance considerably improves. The CA-ABP FER performance loses approximately 0.1dB for every two CRC bits reduced.

Figure 5 shows the performance comparisons of two decoding algorithms for PAC codes with code rate R=40/128 and code length N=128. CA-ABP($T_{I}$, $T_{O}$, $\tilde{L}$) represents that the maximum numbers of inner iteration, outer iteration, and the length of participating CRC bits are $T_{I}$, $T_{O}$, and $\tilde{L}$, respectively. Obviously, the gain of CA-ABP(20, 10, 16) decoding exceeds 0.6 dB compared to that of the CA-List(32) algorithm, but its UER is relatively high. When we further increase the number of iterations, the performance gain of CA-ABP(20, 40, 16) is close to 1.2 dB and 0.2 dB over those of CA-List(32) and CA-List(1024) decodings respectively, but it only leads to a slight increase in UER, which is shown in Figure 5 by the dashed line. In practical applications, we can simultaneously optimize the length of participating CRC bits and the damping factor according to the simulation results of FER and UER.

From the simulation results in Figure 6, the CA-ABP(40, 40, 16) outperforms the state-of-the-art CA-List(2048) decoding, where the gap between them is about 0.2 dB. Moreover, the average complexity of the CA-ABP(40, 40, 16) is much lower than that of the CA-List(2048) algorithm according the analysis of Figure 2.

Furthermore, we also give the relevant simulation results of PAC codes with other code lengths and code rates. Figure 7 shows the decoding results of PAC codes with longer length. It can be seen that compared to the short PAC codes used in Figure 5 and Figure 6, the gain of CA-ABP (20, 10, 16) is narrowed to only 0.1 dB over CA-List (32) decoding. With the code length increasing, our proposed algorithm and the common CA-List decoding reach almost the same performance. At the moment, we can appropriately increase the number of the CRC bits assisting in CA-ABP decoding for performance improvement.

Now, we explore the CA-ABP decoding of punctured PAC codes. We simulate the performance of CRC-PAC codes with the punctured method mentioned by the previous section. We puncture 8, 16, and 24 bits of PAC (128, 96) code, respectively, to analyze the performance loss caused by progressive puncturing. We choose CA-List (32) and CA-ABP (20, 20, 16) as the decoding algorithms of these punctured PAC codes, respectively. As can be seen from Figure 8, for PAC (128, 96) code, the performance gap between the CA-List decoding and the CA-ABP decoding is only 0.6 dB. However, as the number of punctured bits increases, the performance gap between the two decoding algorithms gradually widens. For the punctured PAC (104, 96) code, the performance gap between CA-list and CA-ABP exceeds 1.7 dB. The simulation results fully demonstrate that our proposed algorithm can help to alleviate performance loss the performance loss caused by puncturing. Figure 9 shows the decoding performance of the punctured PAC (112, 96) code after puncturing 16 bits. Obviously, the CA-ABP (20, 10, 16) decoding noticeably outperforms the CA-List (32) decoding, and the gap between them is substantially larger compared with that between the decodings of the original PAC (128, 96) code without puncturing. Also, the UER performance degrades. Figure 10 shows the results of the punctured PAC (224, 192) code with puncturing 32 bits. Compared to the Figure 7, the CA-ABP (20, 10, 16) achieves clearly better performance than CA-List (32) decoding, which means that the BP-like algorithms performs much better than list decodings for punctured PAC codes.

## 6. Conclusions

In this paper, we firstly derived the joint parity-check matrices of CRC-PAC codes suitable for the BP decoding. Then, the CA-ABP decoding algorithm for CRC-PAC codes was proposed, which can effectively enhance the error performance of short blocklength CRC-PAC codes. With the assistance of partial CRC bits, our proposed decoding scheme provided significant performance gain compared with the CA-List decoding. Although the error detection ability is markedly reduced by using more CRC redundant bits in error correction, the UER of the proposed scheme can still be well controlled at a sufficiently low level by adjusting the damping factor or the length of partial CRC bits participating in decoding. The simulation results show that when partial CRC bits are used to improve performance, the codes with higher code rates and shorter code lengths obtain greater performance gain. Therefore, the CA-ABP algorithm is a more effective supplement to short PAC codes, which is obviously beneficial for further 5G evolution and 6G.

## Figures and Tables

**Figure 1 entropy-24-01170-f001:**
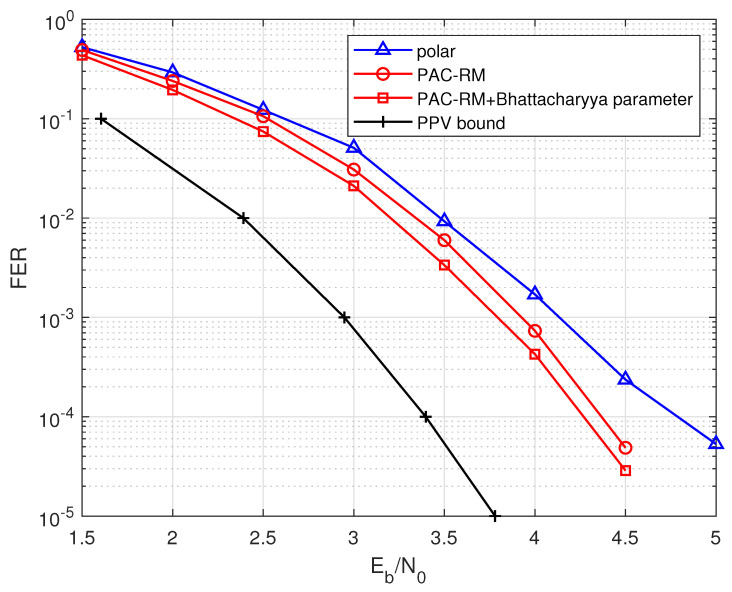
The performance comparisons of the PAC code and the polar code with R=96/128 and N=128.

**Figure 2 entropy-24-01170-f002:**
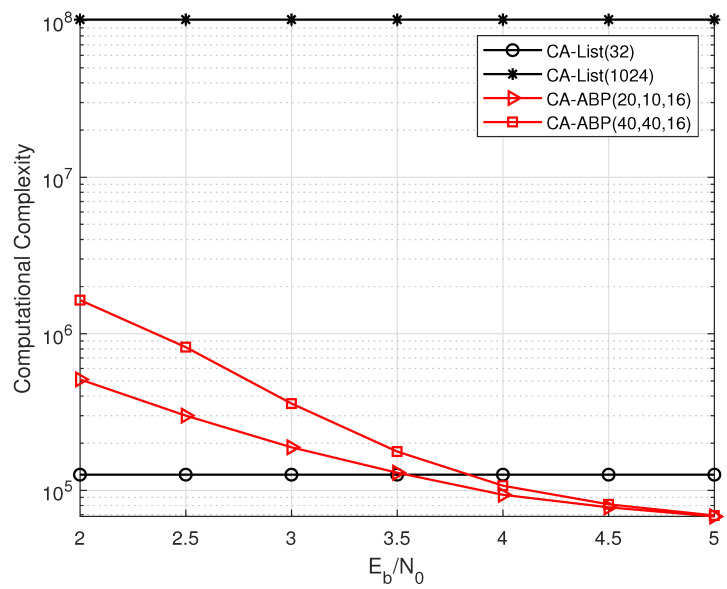
The complexity comparisons of CA-List and CA-ABP on the PAC code with R=40/128, N=128, and L=24.

**Figure 3 entropy-24-01170-f003:**
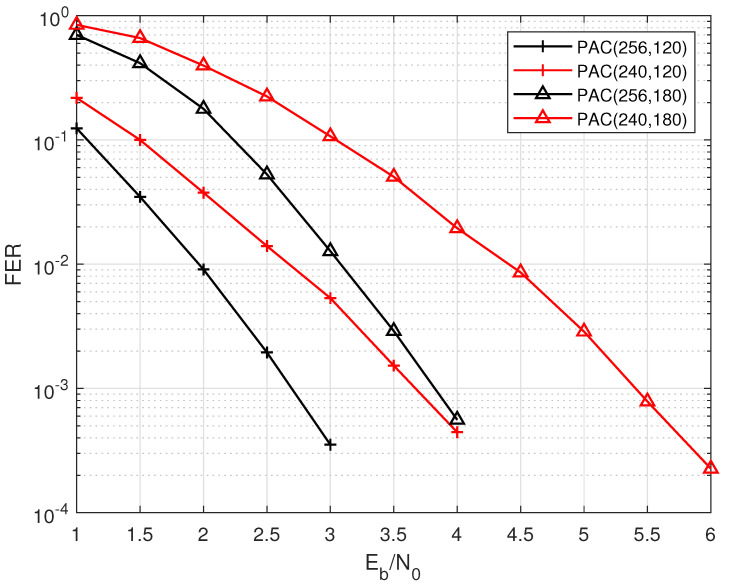
The performance of punctured PAC codes decoded by List(32) for different code rates with N=256 and E=240.

**Figure 4 entropy-24-01170-f004:**
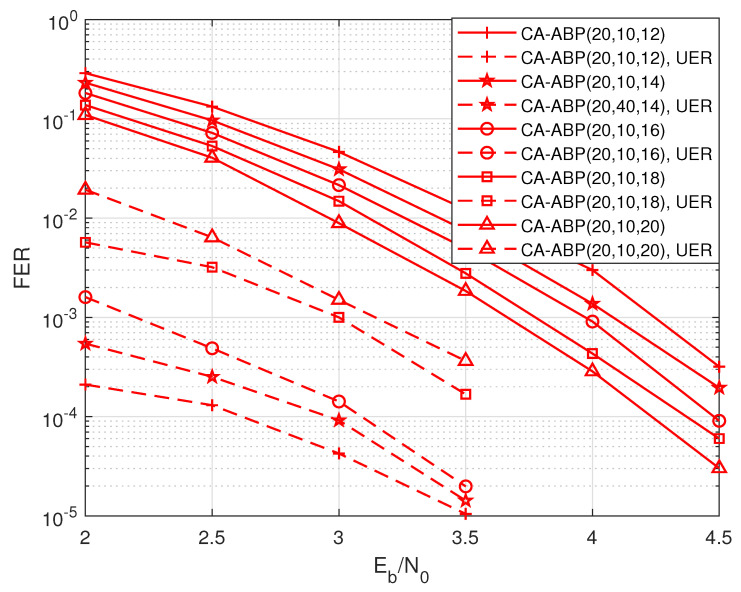
The performance influence of CRC bits on the PAC code with R=72/128, and N=128.

**Figure 5 entropy-24-01170-f005:**
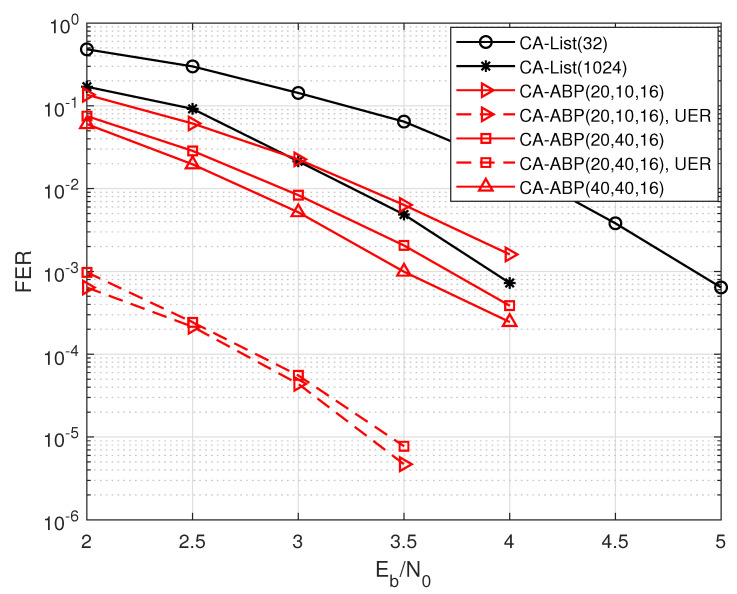
The performance comparisons of the PAC code with R=40/128, and N=128.

**Figure 6 entropy-24-01170-f006:**
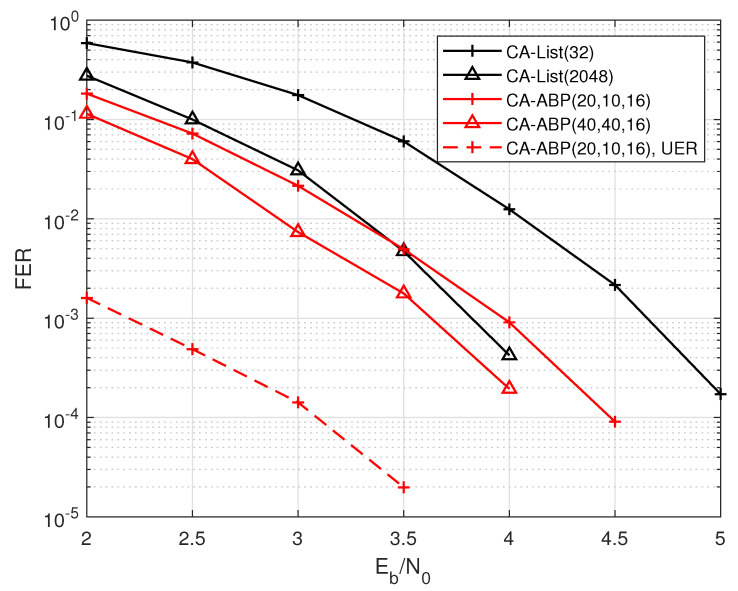
The performance comparisons of the PAC code with R=72/128 and N=128.

**Figure 7 entropy-24-01170-f007:**
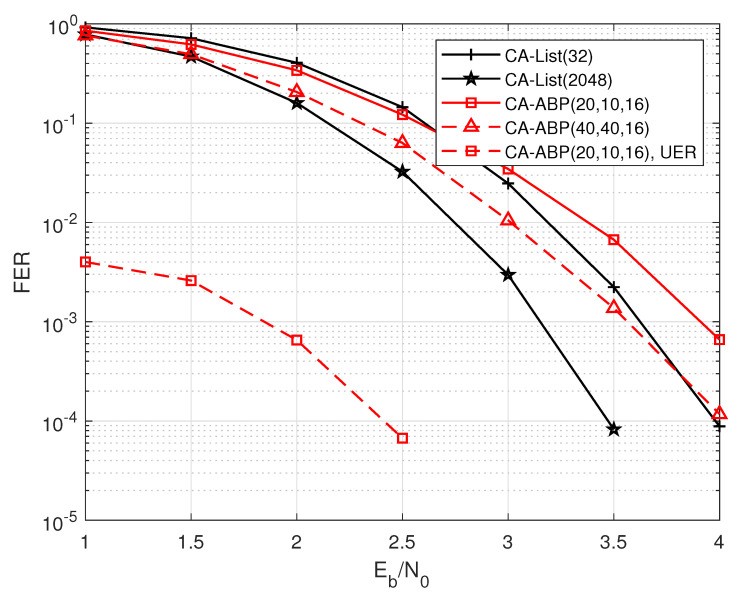
The performance comparisons of the PAC code with R=168/256 and N=256.

**Figure 8 entropy-24-01170-f008:**
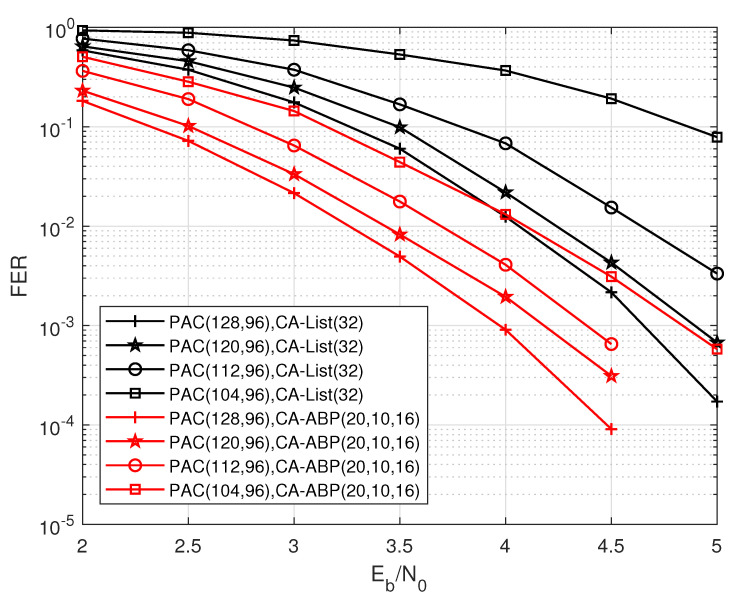
The performance influence of puncturing on the PAC code with R=72/128, N=128.

**Figure 9 entropy-24-01170-f009:**
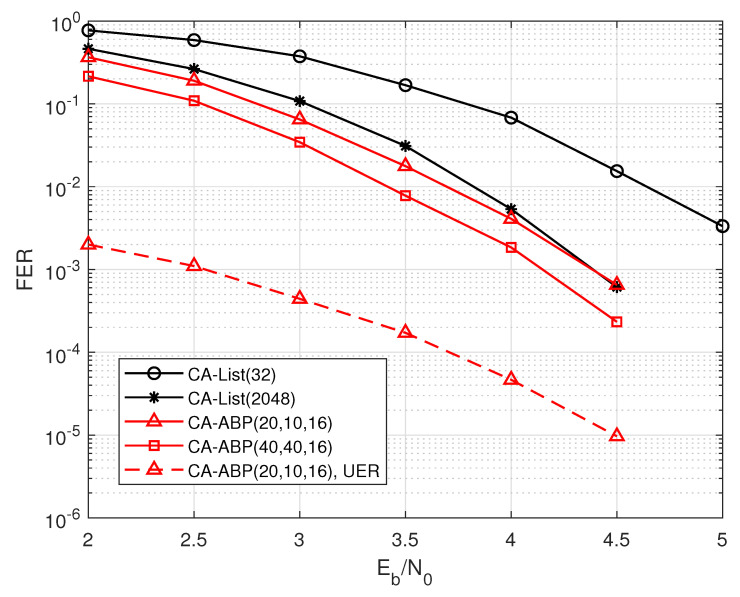
The performance comparisons of the punctured PAC code with R=72/112, N=128, and E=112.

**Figure 10 entropy-24-01170-f010:**
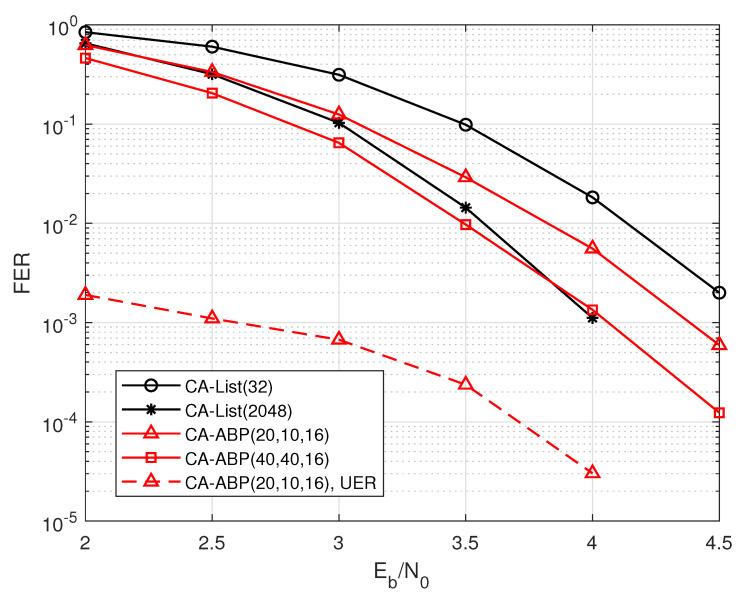
The performance comparisons of the punctured PAC code with R=168/224, N=256, and E=224.

**Table 1 entropy-24-01170-t001:** Complexity comparison of different algorithms for PAC codes.

Algorithms	The Average Computational Complexity
CA-List	$\eta \cdot N\cdot \log_{2}N+K\cdot \left(3\cdot \eta -1\right)\cdot \eta /2$
CA-ABP	$\left(\left(M+\tilde{L}\right)\left(K-\tilde{L}\right)+\left(N-1+K-\epsilon \right)\left(M+\epsilon \right)/2\right)\cdot T_{av}$

## Data Availability

Not applicable.

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
