# Peer review of "CRC-Aided Adaptive BP Decoding of PAC Codes"

_entropy, 2022, doi:10.3390/e24081170_

Round 1
Reviewer 1 Report
This paper considers concatenated CRC-PAC codes and proposes to employ the ABP algorithm for decoding. For this, it identifies the joint parity check matrix of CRC-PAC codes, which also allow to use only a part of the CRC bits. This permits to address the UER criterion in addition to FER.
The paper is clear and well written, but it is not always clear to distinguish what is new from what already exists in the literature. In addition, some more simulation results could help improve the conclusions of the paper. More details about these issues follow.
Major comments
1) It seems that a lot of the components of the paper already exist in the literature: CRC-PAC codes, ABP decoder, etc. As a result, it is unclear whether some parts of the paper are contributions or not. For instance:
- It seems that the ABP decoder introduced in Section 3.2 already exists in the literature, although here it is applied to the CRC-PAC construction. It would be good to mention what is specific to the decoding of this family of codes.
- Section 3.3 provides the complexity of ABP decoder, but again, since this decoder already exists in the literature, it would be good to specify whether this complexity analysis is the same or different from what exists in the literature.
- In the same spirit, line 93 page 4, "We define UER" is a bit strange formulation, since the UER criterion already exists in the literature.
- Section 3.1 provides the parity check matrix of joint CRC-PAC codes. If this is a contribution of the paper, this should be made clearer in the beginning of this Section.
2) Simulation results could be improved. For instance:
- The tradeoff in UER/FER with respect to the number of CRC bits used for decoding could be analyzed more deeply, since this is supposed to be one of the contributions of the paper.
- Given that only PAC codes of length N=2^n can be constructed, the paper proposes to use puncturing so as to consider any code length.This seems prospective, since too aggressive puncturing could severely degrade the code performance. To support this claim, only two cases are considered (puncturing of 16 bits in the code of length 128, and puncturing of 32 bits in the code of length 256). I would suggest to consider more cases.
- It would be good to also compare the performance with respect to other (code,decoder) solutions, like Polar codes with list of ABP decoding.
- Figure 2 only shows three points for UER criterion It would be good to show the results for larger Eb/N0 values, so as to get more points.
Minor comments
- Section 2 is long and addresses several issues. It could be good to cut it into several subsections.
- Page 2, line 47, I would suggest "polynomial complexity" instead of "the polynomial complexity"
- Page 3, before (2), maybe use \llbracket 0,L \rrbracket, instead of i=0,1,\dots,L
- To pass from (7) and (8), it would be good to explicitly say that \phi(mG) = \phi(m)\phi(G)
- Equation (18) uses a \phi, like in (7)/(8). But I guess it is not the same function? It could be use to use a different notation.
Author Response
Thanks to the reviewer for these valuable comments. Point-by-point responses to your comments can be found in the PDF.

Reviewer 2 Report
Paper presents interesting and important results, as a minus I can mentione two things:
1. There are no revieves on the recent works in this area, for an example papers about researches of the implementation of LDPC codes in 5G.
2. Authors need to highlight the benefites of their research in this area.
Author Response
Thanks to the reviewer for your valuable comments. The point-by-point response to your comments can be find in the PDF.

Round 2
Reviewer 1 Report
The authors have correctly addressed all my major comments. Especially, the added simulation results greatly improve the depth of the paper.
I only have some minor comments as follows:
- Page 6, line 147: in the last sentence, I would use something like "for detection, e.g., for lowering the UER criterion" instead of "to detection [...]".
- Page 4, line 113, "According to [16]", instead of "According to the paper [16]"
- When listing the contributions in Introduction, I suggest to always use a direct form (instead of mixing direct forms and indirect ones". For instance, the first point would become : " We derive parity check matrices [...]", and the third point can also be updated accordingly.
- In addition, in this list of contributions, you may want to use \enumerate instead of \itemize so as to show number like what you did in the response to reviewers.
Author Response

(The authors gave the same response as above.)
